# Participatory science methods to monitor water quality and ground truth remote sensing of the Chesapeake Bay

**Patrick Neale**[1]*, **Shelby Brown**[1], **Tara Sill**[1], **Alison Cawood**[1], **Maria Tzortziou**[2], **Jieun Park**[2], **Min-Sun Lee**[2,3], **Beth Paquette**[4]

**1** Smithsonian Institution, Smithsonian Environmental Research Center, Edgewater, Maryland, United States of America, **2** The City College of New York, City University of New York, New York, New York, United States of America, **3** National Institute of Fisheries Science, Busan, South Korea, **4** Independent researcher, Edgewater, Maryland, United States of America

* nealep@si.edu

**Data Availability Statement:** All volunteer collected data is publicly available at serc. fieldscope.org. Spreadsheets with all validation

## Abstract

Measurements by volunteer scientists using participatory science methods in combination with high resolution remote sensing can improve our ability to monitor water quality changes in highly vulnerable and economically valuable nearshore and estuarine habitats. In the Chesapeake Bay (USA), tidal tributaries are a focus of watershed and shoreline management efforts to improve water quality. The Chesapeake Water Watch program seeks to enhance the monitoring of tributaries by developing and testing methods for volunteer scientists to easily measure chlorophyll, turbidity, and colored dissolved organic matter (CDOM) to inform Bay stakeholders and improve algorithms for analogous remote sensing (RS) products. In the program, trained volunteers have measured surface turbidity using a smartphone app, HydroColor, calibrated with a photographer's gray card. In vivo chlorophyll and CDOM fluorescence were assessed in surface samples with hand-held fluorometers (Aquafluor) located at sample processing "hubs" where volunteers drop off samples for same day processing. In validation samples, HydroColor turbidity and Aquafluor in vivo chlorophyll and CDOM fluorescence were linear estimators of standard analytical measures of turbidity, chlorophyll and CDOM, respectively, with $R^2$ values ranging from 0.65 to 0.85. Updates implemented in a new version (v2) of HydroColor improved the precision of estimates. These methods are being used for both repeat sampling at fixed sites of interest and ad-hoc "blitzes" to synoptically sample tributaries all around the Bay in coordination with satellite overpasses. All data is accessible on a public database (serc.fieldscope.org) and can be a resource to monitor long-term trends in the tidal tributaries as well as detect and diagnose causes of events of concern such as algal blooms and storm-induced reductions in water clarity.

data have been archived in the figshare database: https://doi.org/10.25573/serc.26662291.v1.

**Funding:** Support was received from the NASA Citizen Science for Earth Systems Program (https://www.earthdata.nasa.gov/esds/competitive-programs/csesp), Award # 80NSSC22K1912 to PJN, AC & MT. The funder did not play any role in the study design, data collection and analysis, decision to publish, or preparation of the manuscript.

**Competing interests:** The authors have declared that no competing interests exist.

## Introduction

Water quality monitoring programs use a suite of indicators to provide information about overall ecosystem conditions. These include biological indicators like chlorophyll a (Chl), physical indicators like turbidity, and chemical indicators like colored dissolved organic matter (CDOM), which are related to the concentration of algae, suspended particles and dissolved organic substances, respectively, in the water [1]. Each of these indicators can also be related to key water optical properties (such as scattering and absorption) that determine light transmission in aquatic ecosystems, which together are referred to as the Inherent Optical Properties (IOPs). The color of water and its main determinants can also be retrieved from space, using advanced multi- and hyper-spectral ocean color sensors [2,3]. High spatial resolution (e.g., 10–60 m) optical sensors, in particular, can uniquely capture the fine-scale heterogeneities and complexities of inland and coastal aquatic ecosystems [4]. Such sensors include the Landsat-8/9 Operational Land Imager (OLI) (a collaboration between NASA and the U.S. Geological Survey, USGS), the MultiSpectral Instrument (MSI) spectral imager aboard the Sentinel-2A/B missions of the European Space Agency (ESA), as well as NASA's upcoming Surface Biology and Geology (SBG) Designated Observable. These satellite sensors (present and future) provide a unique capability to capture water quality changes in highly vulnerable and economically valuable nearshore and estuarine habitats. Yet, they also require extensive (spatially and temporally) in situ measurements to validate and inform the development of appropriate algorithms to retrieve water quality metrics from satellite data.

The past decade has seen increasing involvement of volunteers in water quality monitoring, but few studies have related ground level data to satellite-based remote sensing of surface water properties [5]. Nevertheless, the potential for volunteer measurements to complement remote sensing observations of the ocean and lakes has been recognized for some time [6]. Two examples of projects harnessing this potential are the SeaHawk/HawkEye project (https://coast-lab.org/hawkeyeCitSci/), and Eye on Water project [7], where volunteers are collecting measurements that can be compared to products derived from remote sensing of ocean color. While these projects are sometimes described as citizen science, community science or volunteer monitoring, we have adopted the more inclusive term, "participatory science" [8,9]. Apart from the above cited examples, most participatory science water quality projects have focused on involving volunteers to sample, and conduct simple physical, chemical and biological assays [1]. On the other hand, there are many participatory science projects that assist in interpretation of remote sensing of air-quality or land features and hazards (e.g. EPA's Air Sensor Toolbox and NASA's Landslide Reporter and Mountain Rain or Snow programs).

The Chesapeake Bay is one of the largest estuaries in North America, located on the East Coast of the United States. The Chesapeake Bay stretches for about 320 km from the northernmost point of the Susquehanna River in Maryland to the southernmost point of the Atlantic Ocean. As a drowned river valley, a notable feature of the bay is its complex shoreline, with many small embayments and rivers that are tidal tributaries in their lower reaches. These shallow areas are a large portion of Bay surface area and important transitional zones in between the watershed and the main bay. They perform many important ecological functions as hot spots for the processing of terrestrial inputs to the Bay, fisheries habitat and refuges for juvenile fish, habitat for shore birds, and many more [10]. Improved water quality, and in particular better water clarity, in shallow waters has been identified as a high priority by the regional partnership responsible for Bay restoration, the Chesapeake Bay Program [11]. Despite their importance, only a few of the tidal tributary areas are subject to high resolution monitoring that includes Chl, turbidity, and CDOM (e.g. Rhode River [12], and Patuxent River [13]). This

contrasts with the extensive, long-term monitoring of the main stem of the Chesapeake Bay which has informed quantitative targets for acceptable water quality [14].

In the Chesapeake Bay, participatory science and non-profit organization monitoring of water clarity has primarily used the Secchi disk. This is a flat disk (~30 cm diameter) that is either all white or has black and white sectors. The disk is lowered into the water and the observer records the first depth at which they can no longer visually distinguish the outline of the disk or the contrast between the white and black sectors. Due to its simplicity of deployment, there are long term records of Secchi disk depths throughout the Bay recorded by a wide variety of observers [15,16]. An extension of the Secchi depth is to compare the observed color of the disk at half depth to a color scale (Forel-Ule scale) which is a qualitative indicator of water quality [17]. Despite its benefits, two key disadvantages of the Secchi disk depth and color scale metrics are that they are subjective measures dependent on observer vision and that the depth and/or scale value reflects the combined effect of all properties that affect light transmission [18]. Thus, there is a need for volunteer methods that more specifically show the contributions of individual properties such as Chl, turbidity, and CDOM to water clarity and can be directly compared to similar satellite data products.

Fluorescence measurements are appropriate for participatory science because there are easy to use instruments such as the Turner Designs Aquafluor, many types of measurements can be made directly on the sample without preparation, and results are immediate. Simple fluorometers can even be constructed as DIY (Do-It-Yourself) projects [7]. Microalgae can be detected via the in vivo fluorescence emitted by the photosynthetic pigment, Chl. The fluorescence emission is dependent on both the concentration of pigment and the physiological state of the cell, so the proportion between the signal and an analytical measurement of Chl can vary widely [19]. Moreover, there can be optical interference from co-occurring turbidity and CDOM [20]. Thus, the few participatory science projects that monitor algae fluorometrically have just taken the measurement as a relative indicator without any specific defined relationship to Chl (e.g. community science lab, https://cesh.bard.edu/csl-alt/). Other properties that can be measured via fluorescence include the cyanobacteria-specific pigments, phycocyanin and phycoerythrin, and CDOM. These moieties all have specific excitation emission properties that allow reliable detection, but with varying quantum yield, i.e. the proportion of fluorescence emitted per photon absorbed. Thus, using fluorescence to make quantitative estimates requires detailed validation of the measurement in the target system.

Cameras in mobile devices (aka "smartphones") can also be used in various ways to monitor surface water properties. Smartphone software packages ("apps") have been developed that use camera imagery to assess water color (EyeOnWater) or surface reflectance (HydroColor). EyeOnWater enables users to classify the surface color of a water body on the Forel-Ule scale and compare current readings with historical readings which have records many decades long in some locations [7]. However, the determinations do not relate directly to IOPs. HydroColor references RGB (red, green, blue) images of the water surface and sky to that of a standard 18% reflectance gray card to estimate surface reflectance, from which turbidity can be estimated based on an empirical relationship [21]. In advance of the work presented here, HydroColor has received limited testing in coastal waters [22,23]. A test of HydroColor in 13 Australian lakes and rivers found poor correspondence between the app's estimate of turbidity and standard gravimetric measurements of suspended particulate matter [24].

Water quality monitoring involving volunteers and professionals not only informs local assessments of water quality but also has the potential to provide valuable ground-truth data for validation of satellite remote sensing products that can ultimately complement the on ground monitoring efforts. Here, we report on volunteer measurements relevant to ground-truthing satellite remote sensing of water quality in the Chesapeake Bay. This manuscript

covers the methods used, the data acquired over two years of deployment, and shows that after reparameterizing on the basis of validation data, volunteer data is suitable for ground truthing remotely-sensed water quality. A future manuscript will focus on the optimization of algorithms used to interpret satellite remote sensing of ocean color in Chesapeake Bay tributaries based on match-ups with volunteer measurements.

## Methods

### Study area and sampling

Sampling was conducted as part of the Chesapeake Water Watch participatory science project, a NASA-funded collaborative effort between the Smithsonian Environmental Research Center (SERC) and the City College of the City University of New York (CCNY). During the period reported on here, November 2021 through December 2023, participants regularly sampled locations in the upper Chesapeake Bay (Fig 1). The study area encompassed the oligohaline to mesohaline region of the Chesapeake Bay (nominal salinity 0.5–18). Some initial testing of methods by project staff occurred prior to this period starting in June 2021, but that data was not included in the analyses reported here. In 2021 and 2022, the sampled locations included the tidal estuarine reaches of the West, Rhode, South, and Severn rivers and adjacent parts of

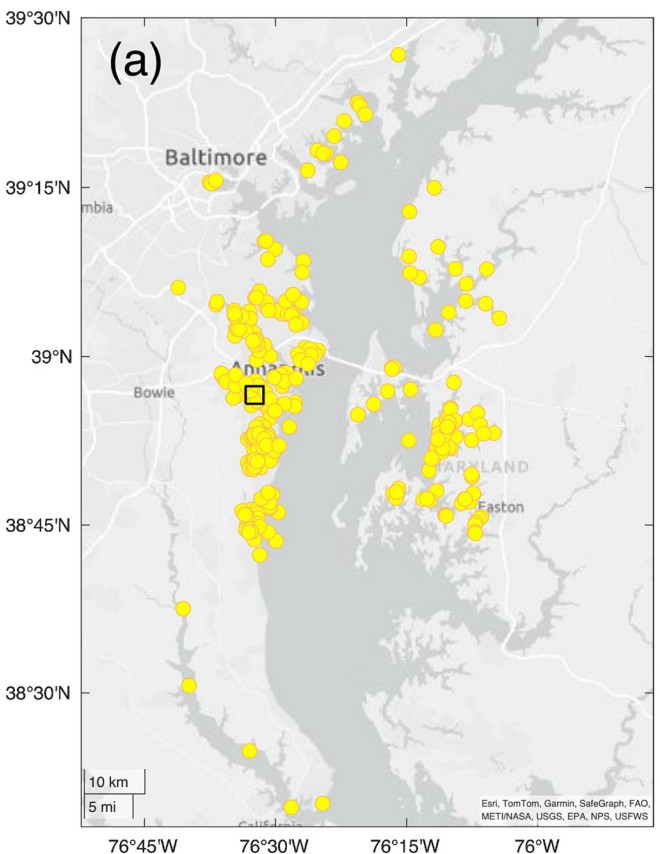

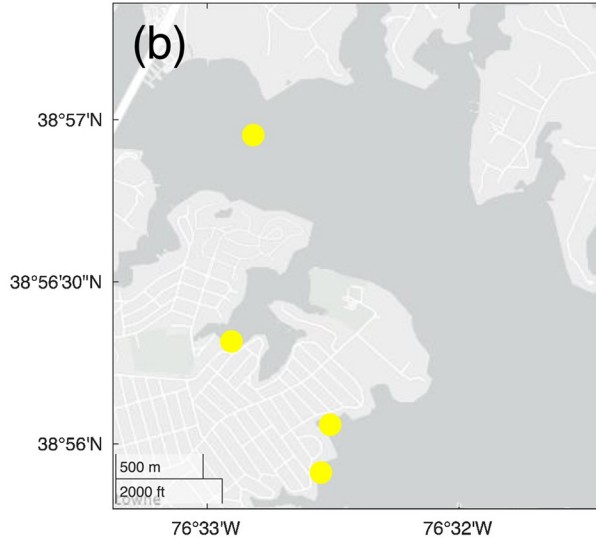

**Fig 1. Chesapeake water watch sampling locations.** (a) Map of the upper Chesapeake Bay in Maryland showing sampling locations in 2021–2023. Four sites in Virginia are not shown in the map. Inset square indicates area of detail shown in (b) a higher resolution map showing 4 sampling locations in the South River near Edgewater, MD. The three nearshore locations are docks sampled regularly by a Chesapeake Water Watch volunteer (co-author Beth Paquette); the mid-channel location nearby is sampled by the Arundel Rivers Federation riverkeeper organization. Basemap content is the intellectual property of Esri and is used herein with permission. Copyright © 2024 Esri and its licensors. All rights reserved.

the main bay. We worked with local Riverkeeper groups–the Arundel Rivers Federation (ARF) and Severn River Association (SRA)—that conducted weekly or biweekly monitoring cruises. Other participants were individuals that made observations and sampled from private or public docks, or private watercraft. No permits were required as sampling took place at sites with public access, or by volunteers sampling on their own property. Project interns and individual volunteers also accompanied SERC staff on monthly sampling cruises on the Rhode River. In total, there were 766 water samples with participation of 57 volunteers during this first period.

In 2023, sampling activities continued in the aforementioned waterways as well as expanding to tidal tributaries on the Eastern shore, north to the Gunpowder watershed, and south to the Patuxent River (Fig 1). These areas were sampled by more individual volunteer participants, additional Riverkeeper groups including the Magothy River Association, Gunpowder Riverkeeper, and ShoreRivers, local high schools and community colleges as well as community groups such as the Advocates for Herring Bay community association. At the end of 2023, there were 2066 water samples and 104 participating volunteers. All volunteers were trained on sampling procedures and measurement protocols using online and/or in person instruction. Measurements are currently ongoing, further expanding the spatiotemporal coverage of our sampling.

As a part of the training, volunteers created accounts on the Chesapeake Water Watch Fieldscope site (serc.fieldscope.org), a data collection and analysis platform managed by Biological Sciences Curriculum Study (BSCS) Science Learning. For each sample, participants were instructed to observe ambient conditions at the measurement site and select appropriate entries on the Fieldscope data entry menu to describe wind speed, surface waves, estimated percent cloud cover, and air temperature (see data form in S1 Table). Although in some cases only HydroColor or surface water samples were collected, in most cases, participants did both. Participants were furnished with small water bottles or 50 mL centrifuge tubes, if these were not available, a used water bottle was acceptable if it previously only contained water. They were instructed to rinse the container three times with water from the sampling location, filling each time with the opening below the surface (to avoid surface scums) before final filling. The containers were stored in the dark at ambient temperature before processing. Samples were left in the dark for a minimum period of 30 min (optimally ≥2 h) to reverse non-photochemical quenching of in vivo chlorophyll fluorescence and (for cold or warm samples) to adjust to room temperature (nominally 22˚C). This timing was based on time course studies that are shown in the results section. Insulated cooler boxes were available at drop-off locations for participants to leave samples.

In addition to locations that were sampled multiple times over the reporting period ("monitoring stations"), we conducted two one-day events that aimed to sample as many locations as possible within the project area (July 14 and October 2, 2023). These synoptic sampling "blitzes", termed "Satellites and Samples" events, were timed to coincide with days that satellite images were scheduled to be acquired by both the OLI (aboard Landsat 8 and 9) and the MSI (aboard Sentinel 2A and 2B) sensors. These events were widely announced via email and open to anyone interested without requiring previous training. Instructions were given on the SERC website that included a template data sheet. Participants were instructed to take a surface sample using a clean plastic bottle (an empty water bottle was recommended), fill out the sheet with a standard description of site conditions, the time and location of the sample, and drop off the sample to one of several processing "hubs" distributed through the area. The hubs were staffed with registered volunteers who processed the samples with project instruments (description follows in "Participatory Science Measurements") and entered the data into Fieldscope. These events contributed 106 and 59 samples on July 14 and October 2, respectively (included in the above total).

## Participatory science measurements

**HydroColor measurements.** Volunteers conducted on-site measurements of surface turbidity using the HydroColor smartphone app. Developed at the University of Maine, HydroColor is a free app for Android and iOS operating systems that enables volunteers to make observations of surface water quality based on water reflectance measurements with a smartphone camera. HydroColor uses the RGB channels of smartphone images taken of a gray card, sky and water surface to calculate remote sensing reflectance, $R_{rs}(R\ G\ B)$ [21]. In addition to measuring Rrs, HydroColor geolocates and timestamps the observation. The developers derived an empirical relationship between red channel reflectance and turbidity which is similar to the algorithm used to derive suspended particulate matter (SPM) estimates from satellite-retrieved $Rrs$ [25]. This relationship assumes a 1:1 proportion between turbidity in NTU and SPM in mg $L^{-1}$.

Chesapeake Water Watch volunteers used two versions of the HydroColor app: version 1 for 2021–2022 measurements and version 2 for 2023 measurements. Version 1 was the version tested by Leeuw and Boss [21], this was updated to version 2 in March 2023. The earlier version used compressed image data from the smartphone camera, this was upgraded in version 2 to use uncompressed ("RAW") image data with expectation that this will give more consistent results between different models of smartphones [26], which we tested in our study. Further information on the developer's pre-release testing and observations on user experiences with the version 2 are reported in (S1 File).

**Aquafluor measurements.** Participants used the Aquafluor handheld fluorometer (Turner Designs, San Jose, CA) to measure in vivo Chl (IVChl) and CDOM fluorescence. It is a simple to use, battery operated instrument suitable for easy and fast measurements in highly variable waters such as the Chesapeake Bay. The IVChl channel uses a 460 nm LED source, a broad band excitation filter with 395 nm center wavelength (CWL) and 130 nm (FWHM) bandwidth, and a 660 nm cutoff longpass emission filter. The CDOM channel uses a 375 nm LED source, a 350/80 nm (CWL/FWHM) excitation filter and a 470/60 nm emission filter. The CDOM channel emission filter is a custom component that Turner Designs configured for Chesapeake Water Watch, as the "stock" emission filter for CDOM measurements is a longpass filter that does not exclude in vivo Chl fluorescence. The channels were calibrated using a modification of the procedures used for sensors in water quality sondes, such as the YSI EXO2 [27]. In vivo Chl fluorescence was calibrated with a 0.625 mg $L^{-1}$ rhodamine WT (Kingscote Chemicals, Miamisburg, OH) solution which is assigned a reading of 66 μg $L^{-1}$ rhodamine-equivalent (RE, temperature corrected to 22˚C). Although its optical properties differ from the desired primary standard, in vivo chlorophyll, rhodamine has a stable, fixed response that is easy to replicate for calibration purposes. Validation measurements are used to determine the relationship of RE measurements to Chl in "live" algal cells. CDOM fluorescence was calibrated with 300 μg $L^{-1}$ quinine sulfate (QS, Sigma-Aldrich) in 0.1N sulfuric acid, with reading assigned a calibration value of 300 μg $L^{-1}$ QS-equivalents (QSE, temperature corrected to 22˚C). QS fluorescence is used as a proxy for natural CDOM and, as for IVChl, the relationship of QSE to CDOM absorbance was determined using validation measurements. Each measurement was conducted in triplicate with ~3–4 mL sample in an acrylic cuvette, reading successively the in vivo Chl, then CDOM fluorescence. These readings are assigned units of μg $L^{-1}$ of RE and QSE, respectively.

Volunteer data for this report was acquired with six identically calibrated instruments. Post-calibration, test samples from the Rhode River tidal tributary at SERC read on all five instruments were within 0 to 10% (mean +5%) for in vivo Chl fluorescence and within -6 to 0% (mean -3%) for CDOM fluorescence of the readings on the reference instrument used to make comparisons in the validation studies (S2 Table).

## Laboratory validation measurements

**Turbidity measurements.** In most cases (>80%) when a surface water sample was taken, turbidity was measured on a benchtop turbidimeter, specifically the AQUAfast AQ3010 Turbidity Meter (Thermo Scientific Orion, Waltham, MA). These instruments are simple to operate and volunteer participants are easily trained to follow a standard protocol. The instrument was checked daily before use that readings were within 1 NTU of a 20 NTU standard and deionized water for 0 NTU. Instruments rarely failed the check, which if it happened was usually due to contamination on the sample vial. Each measurement used 10 mL of sample water in manufacturer provided vials, with readings conducted in triplicate. Instrument calibration was checked yearly and recalibrated as needed with manufacturer furnished turbidity standards.

**Chlorophyll-a measurements.** For selected cases (~15% of total sampling events), participants were requested to sample an extra volume ($\geq$ 250 mL) so that water could be filtered for analytical Chl and CDOM determinations. Sample selection was mainly driven by the logistical concerns of timely transport and processing, but care was taken to acquire samples from all water bodies and in all seasons. As these measurements are more technically demanding, they were only performed by trained project staff. The Chl assay followed the EPA protocol [28]. Triplicate sample aliquots of ~50 mL were filtered on 25 mm glass-fiber filters (GF/F, Whatman Inc.) at low vacuum and stored frozen at -20˚C until analysis (typically within one month). Chlorophyll was extracted with 90% acetone overnight at -20˚C, then the extract was warmed in the dark to room temperature just before analysis. Chlorophyll fluorescence was measured before and after acidification on a Turner 10-AU fluorometer. The values were converted to Chl and pheophytin *a*. The fluorometer was calibrated yearly with Chl standard from Sigma Chemicals.

**CDOM measurements.** For CDOM absorbance, samples were filtered through Whatman GDX syringe filters which have a combination of glass fiber and membrane filters with a nominal 0.45 μm pore size. Filtrates were stored in the dark at 4˚C until analysis, typically within one week. We performed absorbance scans at 2 nm intervals (270–750 nm) using a Thermo Scientific Evolution 220 UV-Vis spectrophotometer. CDOM absorption was measured following the protocol in Tzortziou et al. [29]. Optical density (OD) in 1 cm cuvettes was converted to absorption coefficient ($a_{\text{CDOM}}$ m$^{-1}$) at wavelength, λ (nm), using the equation:

$$a_{CDOM}\lambda = 2.303\frac{OD}{l_g} \qquad (1)$$

Where $l_g$ is the pathlength = 0.01 m.

## Statistical analysis

As a measure of analytical variability for volunteer methods, we calculated the Mean Absolute Deviation (MAD) for specified sets of $n$ triplicate observations (HydroColor turbidity, Aquafluor, IVChl and CDOM fluorescence) using the equation:

$$MAD = \frac{\sum_{i=1}^{n}\sum_{k=1}^{3}abs(x_{ik} - \bar{x}_i)}{3n} \qquad (2)$$

Where $x_{ik}$ are triplicate observations, $k$ = 1 to 3, of the $i^{\text{th}}$ sample with mean $\bar{x}_i$. The efficacy of volunteer measurements as estimators of the validation measurements was assessed using the root mean square error (RMSE) and median % error based on the deviations from the fitted linear regression line. In cases where the intercept was not significantly different from

zero, a regression with no intercept was used. Linear regressions and associated statistics were calculated using the MATLAB *fitlm* function.

# Results

## Participatory scientist measurements

Table 1 summarizes the statistics for each of the variables measured by participants. In general, the sampled locations had moderate turbidity, were mesotrophic and had appreciable concentrations of CDOM. Samples acquired for validation had similar characteristics as the whole data set, shifted to slightly higher values (Table 1). The tidal tributaries which are the focus of this study are dynamic environments due to the varying influence of weather, wind forced flows, nutrient inputs from the watershed and commercial and recreational fisheries [30]. The extent of these dynamics is evident in records compiled by volunteers that regularly sample on a weekly or high frequency basis. Two years of measurements taken by a Chesapeake Water Watch volunteer (co-author Beth Paquette) at three docks on the South River (locations in Fig 1B) show variability at yearly, monthly, and even weekly time scales (Fig 2). For comparison, Fig 2 also shows measurements of the same variables using project methods at a nearby, mid-channel site (location shown in Fig 1B) made by the Arundel Rivers Federation (ARF) river-keeper. For logistical reasons, these samples are only taken mid-April through mid-October. Volunteer and ARF samples are comparable in this period, but the bi-weekly sampling by ARF misses much of the short-term and some of the seasonal variation in this system. Measurements at the three docks were generally consistent, except for April 6, 2022 when there was a high accumulation of tree pollen at one site which lead to an unusually high CDOM reading (Fig 2B) compared to the other sites.

As an example of results of the coordinated synoptic sampling during a "Satellites and Samples" event, we show data from an event on July 14, 2023 in Fig 3. Evident from this "snapshot" of IVChl, CDOM and turbidity in these locations is the increase in each of these properties with distance upstream away from the main bay. These observations can be compared to retrievals from both the OLI/Landsat 9 and MSI/Sentinel 2A on the same day. The results of these comparisons and other applications of the volunteer data for ground-truthing remote sensing products will be reported in a separate publication (in preparation).

**Table 1. Summary statistics for volunteer and validation data.**

| Variable | Period | Units | Whole Data Set | | | | Validation Data | | | |
|---|---|---|---|---|---|---|---|---|---|---|
| | | | lower quartile | Median | upper quartile | Replicate MAD | lower quartile | Median | upper quartile | Replicate MAD |
| **HydroColor Turbidity** | 2022 v1 | NTU | 3 | 5 | 8 | 0.91 (649)[a] | 4 | 7 | 11 | 0.87 (182)[b] |
| | 2023 v2 | NTU | 4 | 5 | 9 | 0.55 (502) | 5 | 7 | 10 | 0.57 (185) |
| **in vivo Chlorophyll Fluorescence** | 2022–23 | μg L$^{-1}$ RE | 27.84 | 41.20 | 61.51 | 1.36 (1386) | 32.82 | 48.18 | 66.87 | 1.55 (205) |
| **CDOM Fluorescence** | 2022–23 | μg L$^{-1}$ QSE | 23.69 | 27.19 | 33.40 | 0.20 (1409) | 25.28 | 30.30 | 38.32 | 0.27 (204) |
| **Benchtop Turbidity** | 2022–23 | NTU | 4.68 | 7.11 | 10.92 | 0.66 (1153) | | | | |

Statistics for volunteer measurements in the upper Chesapeake Bay for 2022 and 2023, for both the full data set and the samples that have paired validation data.

[a]Total number of sample entries in Fieldscope, with triplicate readings for each sample.

[b]Number of samples in the validation data set.

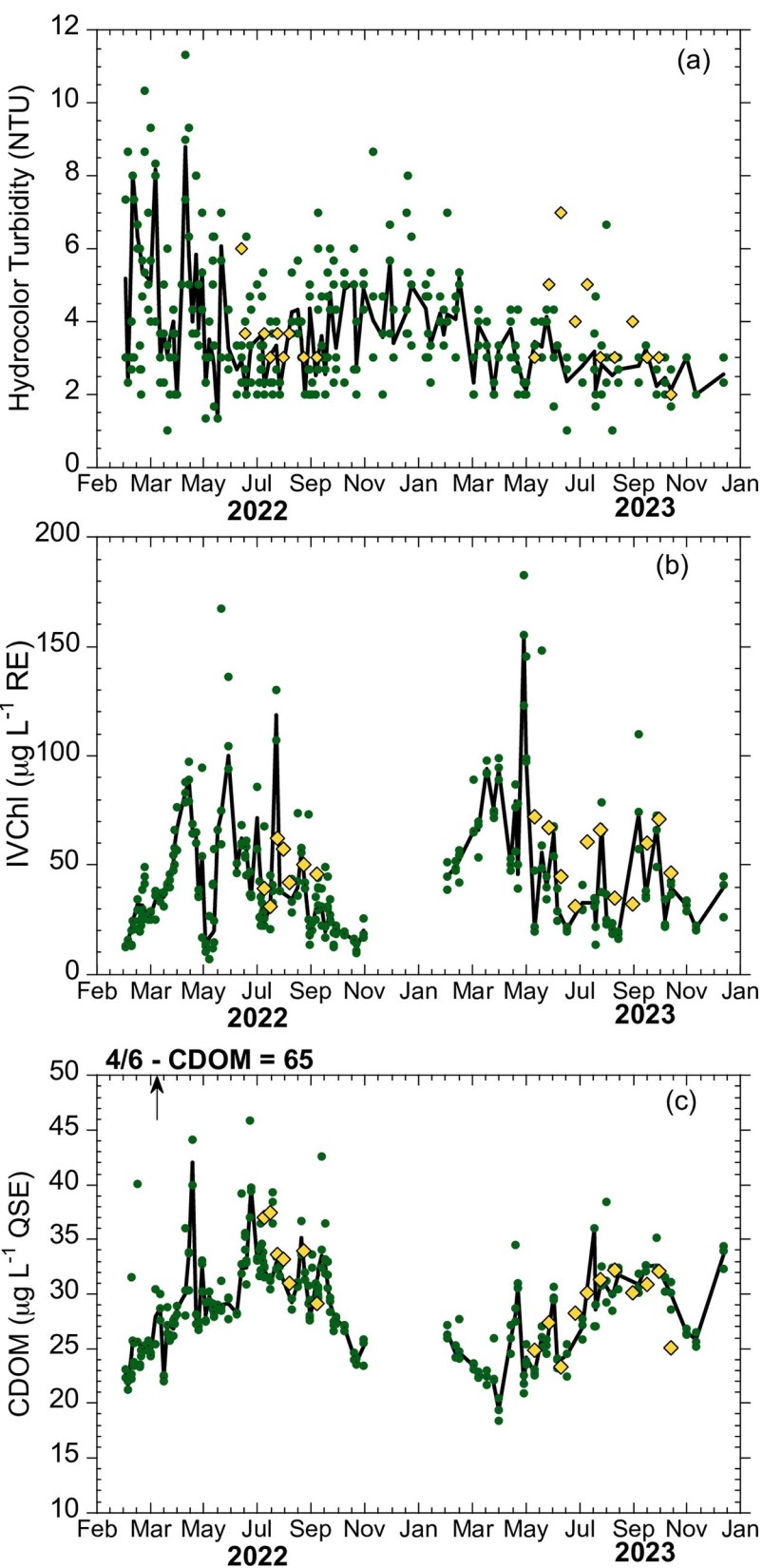

**Fig 2. Examples of participant monitoring measurements.** Observations using project methods taken by a Chesapeake Water Watch volunteer (co-author Beth Paquette) at three docks in the South River near Edgewater, MD (green dots) and an independent set of measurements taken by the Arundel Rivers Federation riverkeeper at a nearby mid-channel site (yellow diamonds, locations shown in Fig 1B). The trends of the aggregate volunteer data (lines) were fit using locally weighted regression (LOESS) with a ~7 day window. (a) turbidity estimated using the HydroColor smartphone app, (b) in vivo chlorophyll fluorescence (IVChl) measured with the Aquafluor, (c) CDOM fluorescence measured with the Aquafluor. The arrow in (c) indicates one volunteer point from April 6, 2022 that was omitted from the plot and trend line fit, on this date CDOM was unusually high (65 µg l⁻¹ QSE) due to high amounts of tree pollen that had entered the water column at one of the docks. Data from the other two docks remain in the plot.

## Participatory scientist measurements–validation

**Turbidity measurements.** HydroColor observations in 2021–2022 were made with app v1 and in 2023 app v2 was used. In each year, approximately 200 of the HydroColor observations were compared with benchtop measurements of turbidity with the AQUAfast on samples taken at the same time and place (Fig 4). The 182 observations by 18 volunteers in 2021–2022 with v1 showed a good linear relationship at a close to unity ratio with benchtop turbidity but also considerable scatter (Fig 4A). The linear regression had a $R^2$ of 0.50 for HydroColor as a predictor of AQUAfast turbidity, a median % error of 33% and the root mean square error (RMSE) of the regression of 5.2 NTU (Table 2). There was a significant intercept in the linear regression of 2.6 NTU, which is less than the RMSE but is necessary for unbiased estimates at low turbidities (residuals are shown in S2 Fig). There were many observations (13% of total) in which the error in the regression estimate was > 7 NTU. This variation was considerably greater than the 24% median percent error for prediction of benchtop turbidity reported by Leeuw and Boss [21] in their developmental tests of HydroColor v1, however those tests were conducted with only three devices (all using iOS operating system). This suggested that at least part of the variability in 2021–2022 could have been due to differences in the response of the phones among the 18 users and motivated the update of the application to obtain greater consistency in the response between phones. The update of app by Thomas Leeuw (S1 File) was completed in February of 2023.

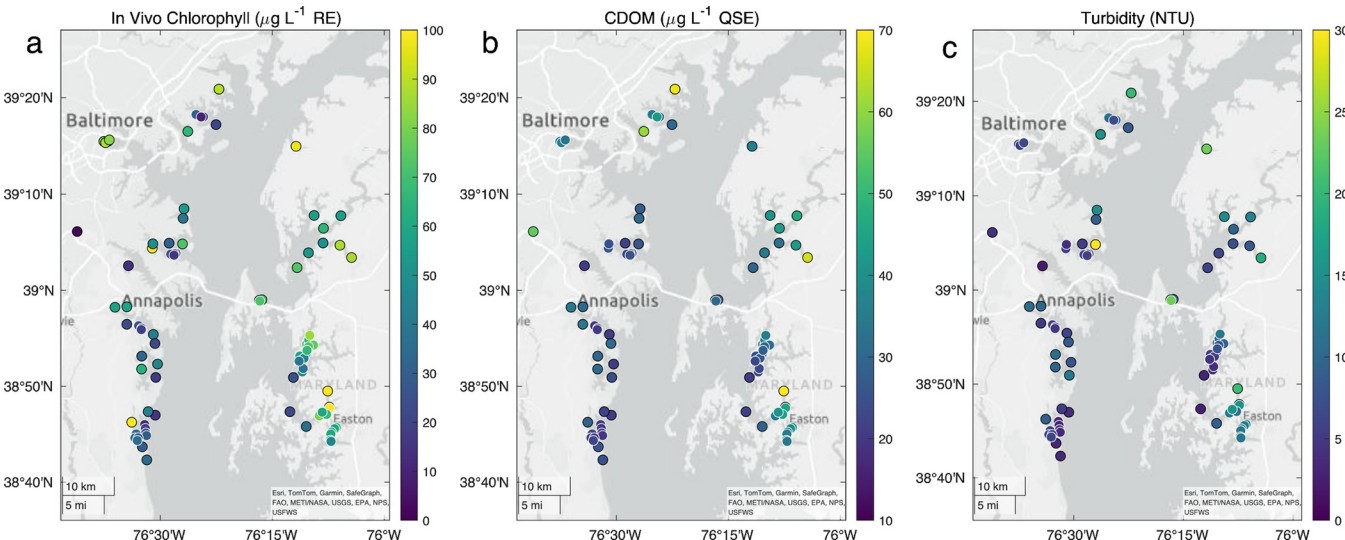

**Fig 3. Maps of the upper Chesapeake Bay showing data from the July 14, 2023 satellite and samples event.** Points show location of samples and color coding on symbols using color bar on right denote values for (a) in vivo chlorophyll fluorescence, (b) CDOM fluorescence and (c) benchtop turbidity measurements. Basemap content is the intellectual property of Esri and is used herein with permission. Copyright © 2024 Esri and its licensors. All rights reserved.

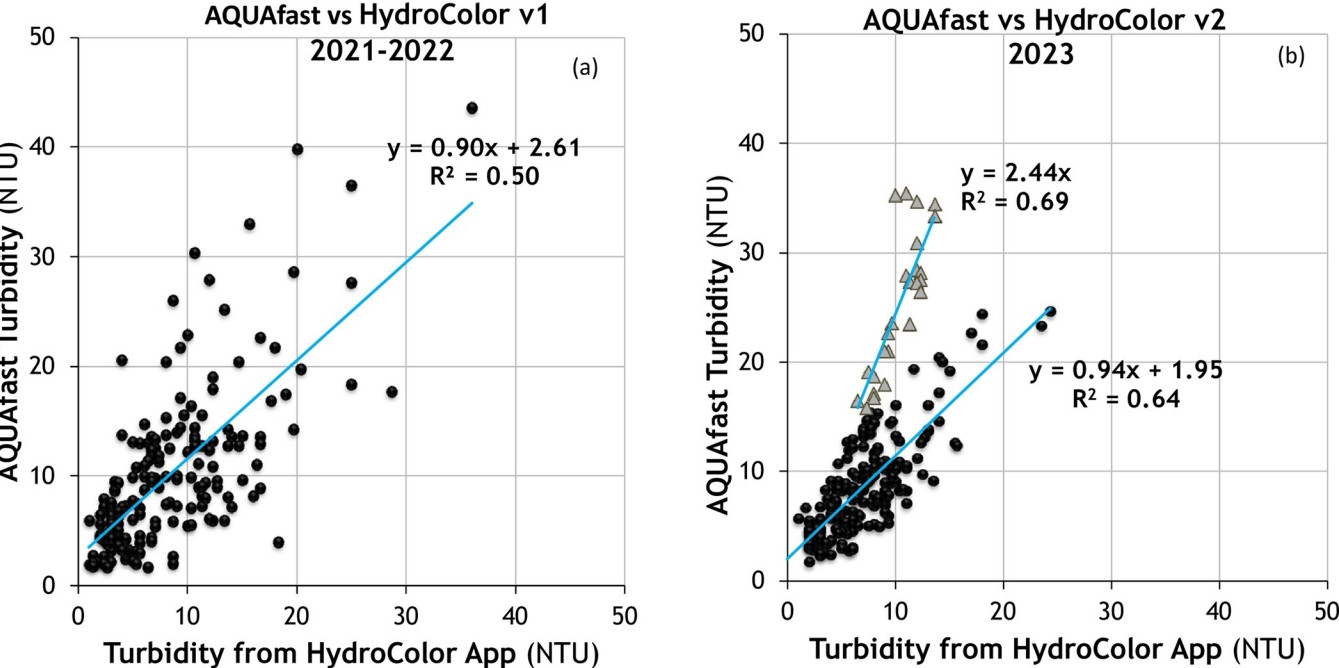

**Fig 4. Comparison of HydroColor estimates of turbidity with benchtop measurements with the AQUAfast.** (a) app v1 used in 2021–2022 and (b) app v2 used in 2023. Gray triangles in (b) are a group of points with high AQUAfast and low HydroColor turbidity (see text). Lines are the fitted linear regressions between benchtop and HydroColor turbidity for each group of samples.

In 2023 using HydroColor v2, most of the observations were also close to unity in the relationship with AQUAfast turbidity (black points, Fig 4B), but with markedly less scatter. The linear regression of these 185 observations by 27 users had an $R^2$ of 0.65, a median percent error of 23% and RMSE of 2.7 NTU (Table 2). Again, an unbiased fit required a small, but significant intercept of 1.9 NTU (residuals in S2 Fig). All average AQUAfast turbidities in 2023 samples were within ±6.5 NTU of the turbidity predicted from the HydroColor v2 regression. Notably, the implementation of v2 improved the overall relationship with an analytical measurement of turbidity even though there were more users contributing to the comparison. The median percent error matched the potential accuracy stated by the developer's (i.e., 24%). However, with the expansion of sampling sites there emerged a group of 25 samples that had a > 2 times greater benchtop turbidity than that measured by HydroColor (gray triangles in

**Table 2. Regressions of validation vs. volunteer data.**

| Variable | Period | Validation Units | Validation Regressions | | |
|---|---|---|---|---|---|
| | | | $R^2$ | Median % error | RMSE |
| HydroColor Turbidity | 2022 v1 | NTU | 0.50 | 33% | 5.2 (182)[a] |
| | 2023 v2 | NTU | 0.65 | 23% | 2.7 (185) |
| in vivo Chlorophyll Fluorescence | 2022–23 | µg L-1 Chl | 0.85 | 16% | 7.0 (202) |
| CDOM fluorescence | 2022–23 | a440 (m-1) | 0.76 | 10% | 0.12 (204) |

Statistics for the regression of validation vs. volunteer measurements in the upper Chesapeake Bay for 2022 and 2023. Analysis excludes outlier group for 2023, (gray points Fig 4B).

[a]Number of paired validation and volunteer measurements.

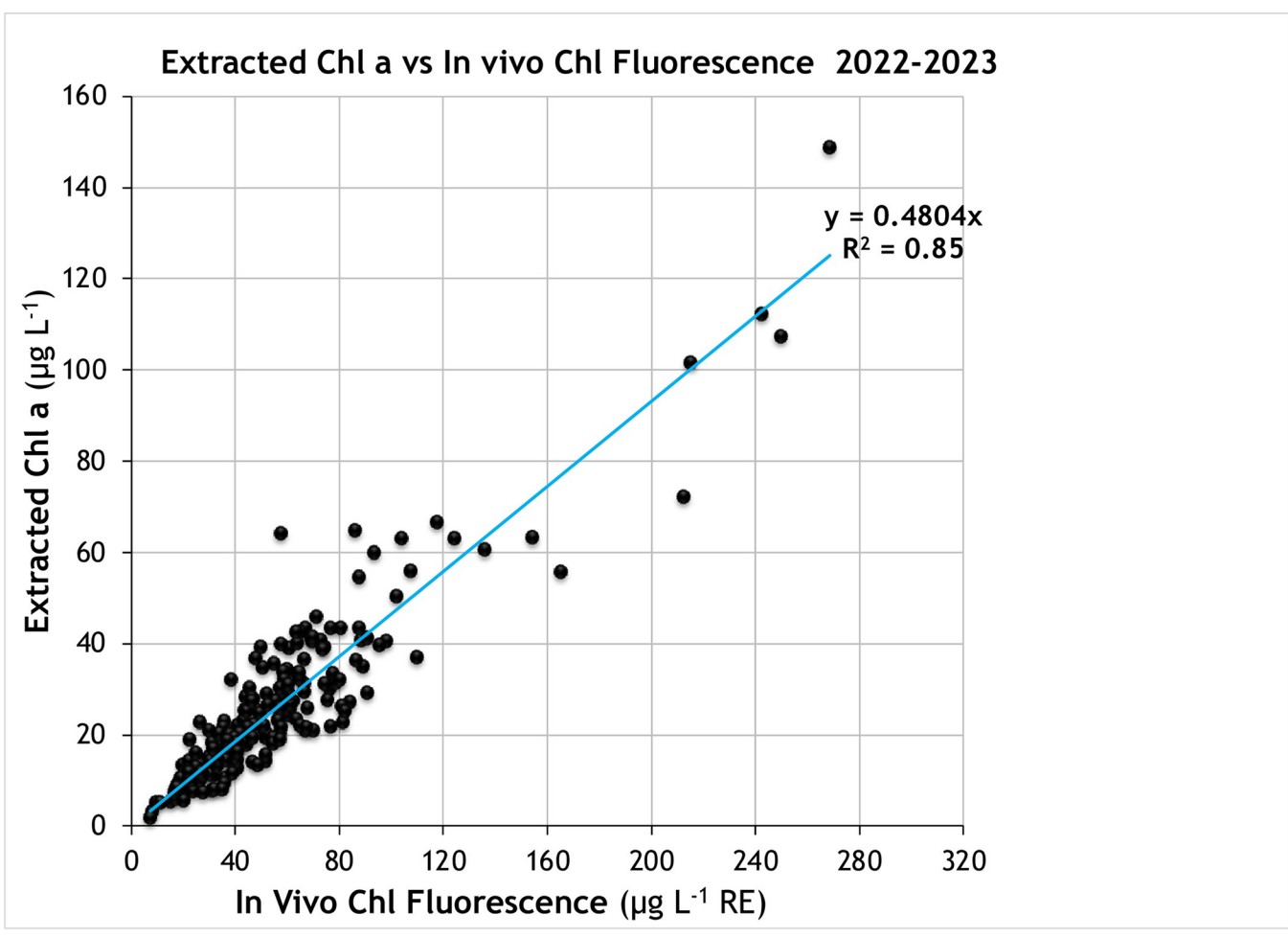

**Fig 5. Relationship of chlorophyll a and in vivo chlorophyll fluorescence in validation samples.** X-axis, IVChl in selected Chesapeake Bay samples measured on the Aquafluor (in rhodamine equivalents, RE, µg L$^{-1}$) vs. y-axis, extracted Chl (µg L$^{-1}$). Complete data set of all validation samples April 2022-October 2023 (n = 202 samples, averages of triplicate measurements). Line shows linear regression assuming a zero intercept.

Fig 4B). The fitted regression to this group did not have a significant intercept. These observations were mainly from tributaries that received treated wastewater (Patuxent and Back rivers) or from shallow areas after wind resuspension events.

**Chlorophyll-a measurements.** The Aquafluor measures IVChl in rhodamine equivalents (RE, µg L$^{-1}$) and validation comparisons were made to establish the relationship with extracted Chl as measured using a standard analytical method. For the samples in the upper Chesapeake Bay, IVChl was a consistent linear estimator of Chl, with Chl averaging about half of the concentration of IVChl rhodamine equivalents (Fig 5). The 202 observations had a linear regression R$^2$ of 0.85 with extracted Chl, a median % error of 16% and RMSE of 7 µg L$^{-1}$ (Table 2). The intercept was omitted as it was not significantly different from zero ($p = 0.65$). Multiple linear regressions were performed that included concurrent measurements of turbidity, CDOM, or both, in addition to IVChl, but adding these variables did not increase R$^2$ (S3 Table). An exception to the general ~0.5:1 relationship of Chl to IVChl were 10 samples from the Severn River during a fall bloom 2021 in which the relationship was closer to 1:1. These samples were encountered in an initial testing period before volunteer sampling began, so are not shown in Fig 5.

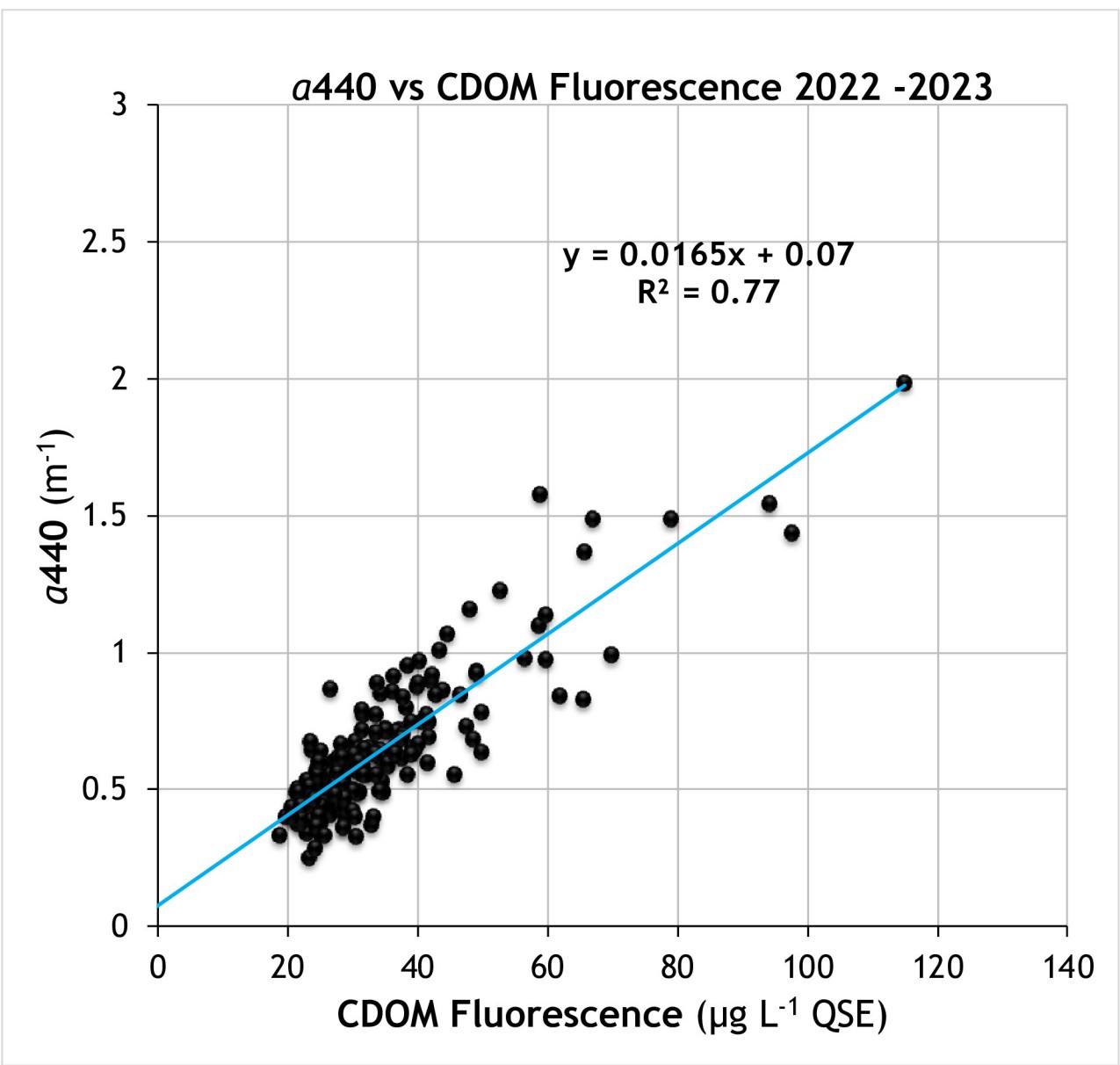

**Fig 6. Validation of Aquafluor CDOM measurement.** Relationship of CDOM fluorescence (µg L$^{-1}$ QSE) to CDOM absorbance measured at 440 nm (m$^{-1}$) in a spectrophotometer (2022 & 2023 data). Line shows fitted linear regression with equation and R$^2$ annotated on the graph.

**CDOM measurements.** For an initial set of validation samples in 2021, the linear regression of CDOM absorbance vs Aquafluor CDOM fluorescence showed an unexpectedly high degree of variability. For example, a regression of 58 samples of absorbance at 440 nm ($a$440, m$^{-1}$) on CDOM fluorescence in June and July 2021 had an R$^2$ of 0.37. This variability was traced to an optical interference from Chl fluorescence which was corrected by installation of custom emission filter that excluded Chl fluorescence (see methods). When these custom machines became available in 2022, we compared CDOM readings in samples that contained both phytoplankton and CDOM vs samples filtered to remove phytoplankton and verified that the interference problem was eliminated by this modified configuration. There was no significant difference in CDOM between unfiltered and filtered samples (see e.g. S1 Fig). A new set

**Table 3. Accuracy and variability of HydroColor for different surface conditions.**

| | V1 | | | V2 | | |
|---|---|---|---|---|---|---|
| Condition | RMSE | Median % error | MAD | RMSE | Median % error | MAD |
| Clear | 4.3 (89)[a] | 31% | 0.49 (251)[b] | 3.1 (81) | 27% | 0.35 (181) |
| Partly Cloudy | 5.9 (64) | 37% | 0.68 (128) | 2.8 (81) | 27% | 0.52 (212) |
| Cloudy | 4.2 (19) | 75% | 0.60 (103) | 3.0 (39) | 26% | 0.97 (32) |
| Calm | 7.7 (44) | 35% | 0.82 (105) | 2.8 (30) | 28% | 0.65 (127) |
| Waves | 4.5 (115) | 28% | 0.75 (393) | 3.3 (166) | 25% | 0.74 (333) |
| Whitecaps | 3.9 (9) | 30% | 0.77 (24) | 3.8 (13) | 33% | 0.90 (18) |

RMSE (root mean square error, NTU) and median % error of benchtop measurements vs turbidity estimated using the HydroColor validation regression (Fig 4), and variability of triplicate HydroColor measurements (MAD, NTU) under different surface observation conditions for v1 (2022) and v2 (2023) of the app. Analysis excludes outlier group for 2023, (gray points Fig 4B).

[a]Number of validation samples for each condition.

[b]Total number of samples for each condition.

of validation samples were acquired during 2022 and 2023. The CDOM fluorescence measured in these samples showed a consistent relationship with CDOM absorbance (Fig 6) and no cross-talk with Chl fluorescence. The 204 validation samples had linear regression $R^2$ of 0.76 for $a$440, a median % error of 10% and RMSE of 0.12 $m^{-1}$ (Table 2). While these instruments are now considered more costly "custom builds", their optical configuration is required to make accurate CDOM fluorescence measurements with unfiltered samples that may contain phytoplankton.

### Participatory scientist measurements–sources of variability

**Turbidity measurements.** Sources of variability in the HydroColor measurement include differences in user proficiency, and variations in cloud coverage and surface wave conditions. We evaluated user proficiency by calculating the MAD for the triplicate entries by each user in the Fieldscope database. Users following the Chesapeake Water Watch protocol took at least three HydroColor measurements and were advised to make as many as two more measurements if the initial three turbidities were not within ±5 NTUs of each other. The user then enters the three readings that were within the acceptable range of one another on Fieldscope. In general, replication was very good, typically there was no more than 1–2 NTU difference between observations. Median variability (MAD per user) was lower in the v2 app, decreasing by 45% from 0.80 NTU for 31 users of v1 to 0.44 NTU for 41 users of v2. For 9 users that made observations using both HydroColor versions, median variability decreased from 0.68 NTU using v1 to 0.44 NTU using v2.

To further explore sources of variability, we evaluated the effect of varying cloud cover and the presence of surface waves. These conditions depart from those assumed to apply in the derivation of equations to correct water leaving radiance for surface reflection [21]. The Hydro-Color algorithm assumes that surface reflection is 2.8% of sky radiance, which is the proportion expected for reflectance of the clear sky from a calm water surface given the viewing position of the camera [31]. The proportional reflectance of sky radiance can be significantly higher under partly cloudy conditions. Likewise, surface waves will result in reflections being viewed at varying angles, potentially increasing the reflectance of sky radiance. To examine how much the presence of clouds and/or surface waves affected the accuracy and repeatability HydroColor observations, we evaluated the RMSE, median percent error, and replicate MAD, as a function of sky and surface wave conditions. The Fieldscope template allows

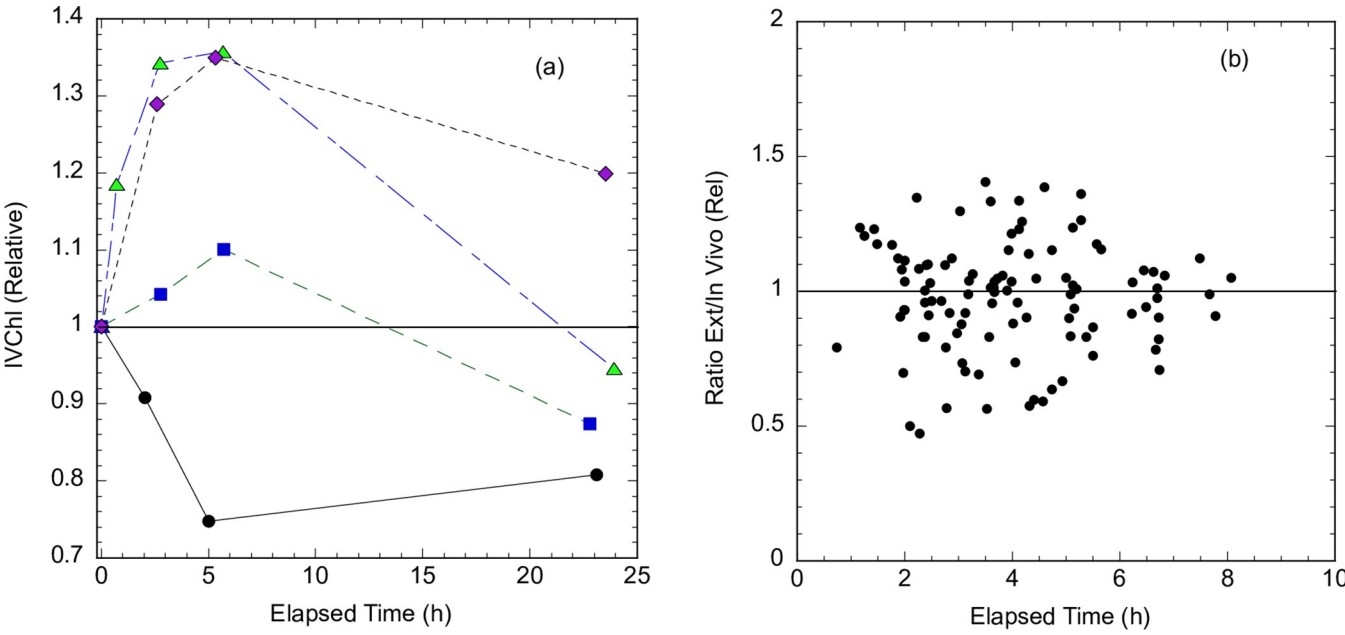

**Fig 7. Effect of dark storage on in vivo Chl fluorescence. (a)** Time courses of fluorescence measured in the Aquafluor for summertime (June-July) samples from the Rhode River, normalized to initial value, the horizontal line indicates a 1:1 ratio. **(b)** Ratio of Extracted Chl to in vivo Chl fluorescence (IVChl) as a function of the elapsed time between sampling and measurement on the Aquafluor. Ratios are normalized to the slope of the linear regression of Extracted Chl vs in vivo Chl fluorescence (Fig 5), the horizontal line indicating a normalized ratio of 1.

volunteers to select five different % cloudiness and surface wave levels (S1 Table). To have a workable sample size for each category, we reduced the classification into three levels of cloudiness: clear (0–5%), partly cloudy (includes mostly clear 25% and partly cloudy 50%) and cloudy (includes most cloudy 75% and cloudy 100%), and three levels of wave action: calm, some waves (rippled to small wavelets and few whitecaps), and whitecaps (frequent and many whitecaps).

There was no indication of greater error relative to turbidimeter estimates of average HydroColor (n = 3) under cloudy conditions vs clear conditions (Table 3). Under all conditions, RMSE was greater than 4 NTU for v1 and decreased to around 3 NTU for v2 (cf. Table 2). Median % error was highest for cloudy conditions in v1. Replicate variability (MAD) was < 1 NTU for all conditions for v1. For v2, median % error did not vary but MAD increased with cloudiness. Surprisingly, there was a trend for lower RMSE with increasing wave action for v1, but that might be confounded with higher between user variability in the "calm" category. The trend was clearer with v2 with increasing RMSE, median % error and MAD as waviness increased. Nevertheless, in all cases the number of samples at the highest level of cloudiness or waviness was limited so the results in those categories should be regarded as tentative. The relative lack of data in these categories is not surprising, since volunteers are cautioned to confirm that conditions are safe before collecting observations.

**Chlorophyll-a measurements.** Although measurement of IVChl has a great advantage of simplicity and speed over the standard analytical practice of measuring Chl in acetone extracts of material concentrated on filters, it is a property of live cells and is also subject to interference from degradation products. Some of the sources of variation cannot be controlled in any practical way, such as species composition, nutrient status, and presence of degradation products such as pheophytin [19,32]. Two factors that can be controlled are temperature and the effect of previous light exposure. Fluorescence yield is inversely related to temperature, the yield

decreasing by about 1.0% per degree C [33]. Thus, all measurements in Fig 5 were made after warming or cooling, as needed, the sample to ambient lab temperature (22˚C). A second source of variation for in vivo fluorescence is the response of photosynthetic organisms to high-light exposure known as non-photochemical quenching [32]. This is especially a concern for surface samples during the summer. The quenching response protects the core complexes involved in photosynthesis, the reaction centers, from damage due to oversaturation with light energy by redirecting much of the energy absorbed by chlorophyll to be released as heat. When quenching is induced, in vivo fluorescence, which mostly originates from reaction center complexes, decreases, even though there is no change in the cellular content of Chl.

To control for the effects of quenching, samples were always placed in the dark at the time of sampling up until measurement, typically a period of > 2 h. This did not typically cause difficulties for participants, as this period is comparable to the time usually required to transport field samples to a hub and arrange for their processing. The period of dark adaptation was guided by the results of several time trials during the summer of 2022, measuring IVChl as soon as possible after sampling and periodically over a period of several hours. In cases when the IVChl increased with time, indicating that the initial sample was quenched, fluorescence increased 10–30% over a period of 2 h then stabilized (Fig 7A). This is consistent with reversal of fluorescence quenching. In one of the trials, fluorescence unexpectedly decreased, but the value stabilized after 5 hours. Fluorescence typically decreased during overnight storage.

To verify that the results of this limited test applied generally to the project sampling, we tracked the time between sampling and measurement on the Aquafluor for 2022–2023 validation samples (data shown in Fig 5). The ratio of extracted Chl to in vivo Chl as a function of dark storage time is shown in Fig 7B. The ratios are normalized to the slope of the linear regression with zero intercept (0.48). Almost all sampling followed the recommended 2 h storage before assay. The normalized extract to in vivo Chl ratio for most of these samples was between 0.85 (lower quartile) and 1.08 (upper quartile). There were a few samples (n = 11) with < 2 h dark adaptation, these samples tended to have slightly elevated ratios (median 1.12), consistent with some residual quenching of fluorescence. Overall, the results support that 2 h dark storage minimizes the effect of quenching on the Aquafluor measurement of in vivo Chl fluorescence.

As the IVChl measurement is performed on a small volume of sample containing the suspended algal cells, there is some inherent variability between replicate measurements. The MAD of a triplicate determination is 1.36 μg $L^{-1}$ (RE), which is 3.3% of the median for the whole data set (Table 1). Settling of aggregates and/or colonial forms is one source of variability. Participant sample processors are advised to gently mix the sample container before withdrawing an aliquot (~2 mL) to load in the cuvette. The measurement also depends on the proximity of cells to the interrogated volume of the sample which is determined by the ~5 mm width of projected LED emission beam. These sources of variability underscore why triplicate determinations improve the accuracy of average IVChl as a linear estimator of Chl (Fig 5).

**CDOM measurements.** As a dissolved, abiotic constituent, CDOM fluorescence measurements have inherently less replicate variability than those of turbidity and IVChl. The MAD for a triplicate measurement with the Aquafluor is 0.20 μg $L^{-1}$ QSE, 0.7% of the median for the whole data set. In addition, CDOM fluorescence measurements were stable for 3 days, if samples were refrigerated (4˚C) then re-warmed to room temperature (22˚C) before measurement (also applies to AQUAfast turbidity). Standardizing measurement at room temperature controls for the temperature dependence in CDOM fluorescence which is about 1% per degree C [34]. Even controlling for temperature effects, there is variability in the relationship between CDOM fluorescence and absorbance, both at 300 nm and 440 nm. This variability occurs because the source of the CDOM and extent of its subsequent photochemical or microbial

transformation in estuarine waters impact, in different ways, its absorption and fluorescence characteristics [e.g., 29,35,36]. These are the main contributors to the 10% median error for using fluorescence as a proxy for CDOM absorbance at 440 nm (Table 2).

## Discussion and conclusions

Based on the 2+ years of sampling by participants in the Chesapeake Water Watch project, we found that volunteer-collected data using the HydroColor app and portable machines consistently estimated variables strongly affecting water clarity in upper Chesapeake Bay tributaries. These approaches are particularly useful for obtaining data, both field and remotely sensed, from the highly dynamic tidal tributaries many of which are severely undersampled for Chl, turbidity, and CDOM. Documenting short-term variation in these systems, especially in the spring, is important to understanding ecosystem functioning and responses to forcing variables [37]. Riverkeepers and other groups actively sample these areas for other variables such as dissolved oxygen, nutrients, and bacteria, while typically the only measure of water clarity is the Secchi depth. Chesapeake Water Watch's methods and instruments expand the volunteers' toolkit for understanding what is affecting water transparency (or lack thereof). Such information is critical in guiding actions to meet water clarity targets for the shallow regions of the Chesapeake. Improved water clarity in these areas is a key part of the Chesapeake Bay Program's efforts for sustaining healthy submerged aquatic vegetation and benthic habitat, limiting algal growth, and improving overall ecosystem functioning [38].

For the tidal tributaries in the upper Chesapeake Bay, we found that the smartphone app HydroColor enabled volunteers to collect observations of turbidity with low replicate variability that were linear predictors of paired measurements with the AQUAfast benchtop turbidimeter. The HydroColor v2 estimates of turbidity were robust to variations in sky and surface wave conditions and were comparable among different users/smartphone models. These factors were all considered possible sources of variability by the App developers and have been cited as explanations for a large degree of scatter in the relationship between HydroColor and independent measurements in previous studies [21,22]. Our results of a good linear relationship between average HydroColor and AQUAfast measurements contrast with the findings of Malthus et al. [24], who found no consistent relationship between HydroColor turbidity and gravimetric SPM assays. There may be several reasons for these contrasting results. Malthus et al. [24] used HydroColor in a wide variety of Australian lakes and rivers, which differed greatly in relative composition of optically active constituents. Our study was limited to the Chesapeake Bay where there is less variation in the types of materials affecting water optics. Also, our study compared HydroColor turbidity to AQUAfast turbidity, not SPM. Because of differences in particle size and shape, the relationship between SPM and turbidity may have differed considerably among the water bodies studied by Malthus et al [24]. In contrast, turbidity and SPM are strongly correlated in the Chesapeake Bay [39]. Finally, the Malthus et al study used HydroColor v1, and our results show that v2 significantly improves the precision of HydroColor estimates of benchtop turbidity by reducing differences in reflectance estimates among smartphone models. Modifications to the user interface facilitating better positioning of the phone and image capture also reduced the variability between replicate observations. While we have not made any systematic comparisons, we expect that HydroColor *Rrs* estimates are closer to those measured using research grade reflectance radiometers. Studies with v1 reported substantial differences between HydroColor and radiometer reflectance [23,24]. However, a recent study of lakes in Hungary which worked directly with smartphone RAW data to derive reflectance in a similar manner as HydroColor v2, found good correspondence between smartphone camera and radiometer-based estimates of RGB *Rrs* [26].

Another result that became evident with the reduced scatter of points in v2 is a group of samples with HydroColor turbidity > 7 NTU for which benchtop (AQUAfast) turbidity was about double (> 15 NTU). At lower turbidities, this group could not be clearly separated from other validation samples, but their presence may have contributed to the positive intercept in the validation regression. Our preliminary observations are that these samples come from tributaries which received treated wastewater (e.g. Patuxent and Back rivers) or from shallow areas after late summer wind resuspension events. The common factor could be a higher percentage of particulate organic material with different scattering properties than usually encountered in coastal marine waters. This would change the relationship between AQUAfast turbidity, which is based on 90° side scatter like most benchtop turbidimeters, and HydroColor turbidity, which is based on backscatter. An increase in the proportion of total scattering to backscatter linked to such an increase in organic content of the Chesapeake Bay particulates has been invoked to explain a long-term increase in total scattering in the bay without an increase in satellite $Rrs$ [40]. Similarly, preliminary results show good agreement between HydroColor turbidity measurements and satellite retrievals of SPM from Landsat 8/9 OLI and Sentinel-2A/2B MSI that, like HydroColor, are based primarily on the backscattering signal (*manuscript*, *in preparation*). We expect that additional observations will assist in better defining conditions when the relationships between AQUAfast and HydroColor turbidity are different. Also planned are additional comparative measurements of turbidity and gravimetric SPM.

Our results support using the Aquafluor to obtain quantitative estimates of Chl and CDOM in the upper Chesapeake Bay which was the focus of project participants in the period 2021–2023. In validation samples, Chl was proportional to IVChl after sufficient dark adaptation. A 2h dark adaptation appears to be required when working with surface samples to consistently reverse the effects of non-photochemical quenching. Shorter periods may be sufficient for incubator-grown laboratory cultures, for example 3 min as suggested by Cremella et al. [20] for the Aquafluor measurements of IVChl. However, our results are consistent with studies using other models of fluorometers that have shown that 1 hour or more adaptation is required for recovery of IVChl from near-surface exposure to solar irradiance [32,41].

Another consideration for IVChl in samples from turbid estuaries, is optical interference from turbidity and CDOM. In near-shore samples of the Rio de la Plata estuary, a cyanobacteria-dominated lake and urban stream in Uruguay, interference from turbidity and CDOM artifactually increased readings of IVChl by the Aquafluor [20]. In contrast, combining IVChl with turbidity or CDOM, or both, as explanatory variables did not improve the prediction of Chl in our validation samples (S3 Table). However, the Uruguayan study worked with samples with much higher turbidity (36 to 156 NTU) and CDOM (1 to 13.6 $a$440 m$^{-1}$) than typically encountered in the upper Chesapeake Bay [20]. At the lower levels in our samples the positive interference to emission in the IVChl channel caused by turbidity scattering and CDOM red fluorescence may be counterbalanced by the negative interference due to the same factors causing attenuation of the excitation irradiance. Additional testing for potential interference effects is advised when initiating volunteer projects using the Aquafluor in other water bodies, and will be continued as the Chesapeake Water Watch expands to other parts of the Chesapeake Bay. Although the Aquafluor model required to accurately measure both Chl and CDOM fluorescence on a single unfiltered sample is presently considered a more costly "custom build" by Turner Designs, our results are the first step towards showing that this, or similar systems [e.g., 7], using the same optical configuration should be a suitable for tandem assay of Chl and CDOM fluorescence across a wide range of environments.

In summary, the tools described in our report enable a more comprehensive monitoring of variables that are key to achieving water clarity goals throughout the Chesapeake Bay. In

particular, these approaches are well suited for improving spatio-temporal coverage of conditions in tidal tributaries, sub-estuaries and other shallow areas in the bay that are often undersampled. Excessive levels of the water clarity variables considered here (Chl, turbidity, CDOM) are indicators of algal blooms, high suspended sediment and urban runoff, respectively, in these areas which are the focus of intensive recreational use, submerged aquatic vegetation, and aquaculture (e.g. oyster farms). These conditions can be detected from both frequent surface samples by volunteers as shown in this report, and from retrieval of these variables from high-resolution remote sensing images. The use of volunteer samples to groundtruth satellite remote-sensing retrieval algorithms is the subject of a future manuscript (in preparation).

## Supporting information

**S1 Table. Listing of the data entry types on the CWW Fieldscope site, serc.fieldscope.org.**
(DOCX)

**S2 Table. Intercomparison of in vivo chlorophyll-a (IVChl) and CDOM fluorescence readings for six identically calibrated Aquafluor units.**
(DOCX)

**S3 Table. Results of multiple linear regression calculations with extracted Chl as the dependent variable and selected volunteer measurements as the independent variables.**
(DOCX)

**S1 Fig. Multiple sample comparison of Aquafluor units SERC1 and SERC2 CDOM and IVChl measurements.**
(DOCX)

**S2 Fig. Plots of the residuals for the validation regressions.**
(PDF)

**S1 File. Developer testing and user experience with HydroColor v2.**
(DOCX)

## Acknowledgments

Most importantly, we acknowledge the hard work and dedication of all our 104+ volunteers, without whom there would not be a Chesapeake Water Watch. Important contributions were also made by the staffs of the Advocates for Herring Bay, Arundel Rivers Federation, Gunpowder Riverkeeper, Marshy Point Nature Center, Severn River Association, ShoreRivers and Magothy River Association. We are also grateful for help from our undergraduate interns Zoe Kehs, Elizabeth Gaudlip, Diana Menendez, Elizabeth Ramos and Samantha Elliott, our high school interns Parker Franco, Lucas Kunkel, Sophia Huang, and Elias Arnold and our teacher interns, Robert Ross and Candace Taylor. Sample processing hub assistance was provided by Tammy Domanski at Anne Arundel Community College, Kathy Gramp at Advocates for Herring Bay, Patterson Clark, and Kate Moore. Rachael Mady provided helpful comments on the draft manuscript.

## Author Contributions

**Conceptualization:** Patrick Neale, Alison Cawood, Maria Tzortziou.

**Data curation:** Patrick Neale, Tara Sill.

**Formal analysis:** Patrick Neale.

**Funding acquisition:** Patrick Neale, Alison Cawood.

**Investigation:** Patrick Neale, Shelby Brown, Tara Sill, Maria Tzortziou, Beth Paquette.

**Methodology:** Patrick Neale, Maria Tzortziou.

**Project administration:** Patrick Neale, Shelby Brown, Tara Sill, Jieun Park, Min-Sun Lee.

**Supervision:** Patrick Neale, Alison Cawood, Maria Tzortziou.

**Visualization:** Tara Sill, Jieun Park.

**Writing – original draft:** Patrick Neale.

**Writing – review & editing:** Patrick Neale, Shelby Brown, Tara Sill, Alison Cawood, Maria Tzortziou, Jieun Park, Min-Sun Lee, Beth Paquette.

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
