## [Decision Letter · Decision Letter 0]

15 Jul 2024

PONE-D-24-22133Participatory science methods to monitor water quality and ground truth remote sensing of the Chesapeake BayPLOS ONE

Dear Dr. Neale,

Thank you for submitting your manuscript to PLOS ONE. After careful consideration, we feel that it has merit but does not fully meet PLOS ONE’s publication criteria as it currently stands. Therefore, we invite you to submit a revised version of the manuscript that addresses the points raised during the review process.

We look forward to receiving your revised manuscript.

Kind regards,

Steven Arthur Loiselle

Academic Editor

PLOS ONE

Journal Requirements:

4. We note that Figures 1 and 3 in your submission contain [map/satellite] images which may be copyrighted. All PLOS content is published under the Creative Commons Attribution License (CC BY 4.0), which means that the manuscript, images, and Supporting Information files will be freely available online, and any third party is permitted to access, download, copy, distribute, and use these materials in any way, even commercially, with proper attribution. For these reasons, we cannot publish previously copyrighted maps or satellite images created using proprietary data, such as Google software (Google Maps, Street View, and Earth). For more information, see our copyright guidelines: http://journals.plos.org/plosone/s/licenses-and-copyright.

a. You may seek permission from the original copyright holder of Figures 1 and 3 to publish the content specifically under the CC BY 4.0 license.  

Natural Earth (public domain): http://www.naturalearthdata.com

Additional Editor Comments:

The manuscript was reviewed by two reviewers, who appreciated the study in this important area, both subject-wise and geographically. However, there were a number of inconsistencies, in particular in the data analysis, that need to be addressed to meet the standards of the journal. In addition, it would be positive to separate the conclusions from the Discussion, and use the Conclusions to make suggestions on how these approaches might be appropriate (or not) for other waterbodies, bother marine and fresh, making the study useful for the wider audience of readers.

Reviewers' comments:

Reviewer's Responses to Questions

**Comments to the Author**

1. Is the manuscript technically sound, and do the data support the conclusions?

Reviewer #1: Partly

Reviewer #2: Yes

2. Has the statistical analysis been performed appropriately and rigorously? 

Reviewer #1: No

Reviewer #2: Yes

3. Have the authors made all data underlying the findings in their manuscript fully available?

Reviewer #1: No

Reviewer #2: Yes

4. Is the manuscript presented in an intelligible fashion and written in standard English?

Reviewer #1: Yes

Reviewer #2: Yes

5. Review Comments to the Author

Reviewer #1: Review: Neale et al., Participatory science methods to monitor water quality and ground truth remote sensing of the Chesapeake Bay, ID PONE-D-24-22133

This manuscript describes a set of participatory (citizen science) measurements of water quality parameters (turbidity, Chl, CDOM) in the Chesapeake Bay area. The low-cost participatory methods are compared to benchmark instruments to determine their accuracy, and replicate measurements are used to estimate the uncertainty.

The manuscript provides an interesting case study in using low-cost methods for water quality measurements, which would be interesting to anyone intending to perform similar measurements or use the resulting data. It is very well-written and easy to read. The discussion section in particular is excellent. However, there are significant shortcomings in the statistical analysis and data visualisation, which are discussed in detail below, along with some minor revisions (mostly typographical). These major issues need to be resolved before the manuscript can be recommended for publication.

Revisions

Abstract

The comparison between HydroColor v1 and v2 is very interesting to anyone who has used the app or is considering doing so in the future, so mentioning it in the abstract would increase the appeal of the manuscript.

Introduction

47, 48: For clarity, please specify that Chl and CDOM are concentrations.

110: “in vivo” is italicised here but not elsewhere.

Methods

Figure 1 is rather low quality/DPI, making it difficult to read. Please replace it with a better quality image.

178: There should be a comma after the closing bracket (typo).

194-199: It would be good to clarify here that the satellite comparison is discussed in an upcoming manuscript, not in the present one. I kept expecting to see it while reading, but it only came back towards the end of the discussion as future work. Similarly, I’m not sure if “and ground truth remote sensing” really belongs in the title of this manuscript.

220: A 1:1 proportion, i.e. 1 NTU = 1 mg/L, or a linear relationship?

226: Reference [38] would be appropriate here, since it was the basis for upgrading HydroColor to RAW imagery. It is very nice to see the authors testing the performance of the new app version in this manuscript.

263-264: What was the result of this daily check? How stable were the instrument readings?

269: How were these cases selected? Could the selection process have introduced a bias in the results?

273: There is a space between the ~ and 50 (typo).

The aggregate RSMD (Equation 2) is not a sufficient description of the uncertainty/variability in the volunteer measurements. It only provides a very rough measure of the variability of one method, but is sensitive to outliers, depends on the unit scale, and cannot be used in further analysis, e.g. weighted regression. It would be better to estimate the uncertainty in each measurement from its replicates (the inner sum in Equation 2) and plot these in the scatter plots (if visually possible) and use them to weight the regression line and statistics. To express the variability of each method, present the distribution of these individual uncertainty estimates, e.g. as a histogram or boxplot, or simply with the me(di)an and some chosen quantiles (1 or 2 σ, 25%-75%, etc.). The me(di)an uncertainty can then be used in the rest of the analysis, e.g. Tables 1 and 2. It would also be very useful to know if the uncertainty depends on the value - the same absolute uncertainty in NTU has a very different significance to a 5 NTU measurement vs. a 30 NTU measurement.

Results

Table 1 shows fluorescence first, then turbidity, while the rest of the manuscript discusses them in the opposite order. It would be more intuitive to keep the order consistent.

Figure 2: The trend line should be removed during the periods without data, approximately November-February in panels (a) and (b). Moreover, the caption should explain what type of average the trend line is and what its window width is.

Figure 2: I’m not sure I understand why the high-CDOM outlier was removed. Doesn’t it further prove the point that short-term variability is very high, thus making a case for participatory monitoring? Or is it because the pollen do not result in actual CDOM in the water, but disturb the measurement? In the latter case, the caption should be changed to clarify this, as it currently suggests the CDOM concentration itself was high. Furthermore, was this effect negligible on all other days?

Figures 2, 4, 5, 6: These figures would benefit from some kind of visual indicator of the uncertainty in the data, so the reader can determine the significance of outliers/differences without having to go back and forth between figures and tables. If the uncertainties differ a lot between points, then consider plotting individual bars; if not, then consider plotting an example in one of the empty corners of each panel.

Figures 2, 4, 5, 6: Additional panels showing the residuals (y-x or regression-value) would make it clear if there are any biases, how the error depends on the mean value, etc.

Figure 3: This has the same image quality/DPI issue as Figure 1.

Figure 3: Please change the colour bar to be more representative of the results and easier to read for those with colour blindness. See [https://doi.org/10.1038/s41467-020-19160-7]. Between this and the low DPI, it is quite difficult to see the suggested upstream/downstream effect. A thin black outline around each point would also help - there are some spots where I’m not sure if I’m seeing a dot or a bend in the coastline.

Figure 4: It would make more sense to have the benchmark measurements on the horizontal axis and the HydroColor measurements on the vertical axis, since the latter is being validated to the former.

Figure 4: Very minor - you could add (v1) and (v2) to the panel titles to make it more immediately obvious what the difference between the two is.

Figure 4: The difference in spread would be more visible if both panels used the same scale/limits on the horizontal and vertical axes.

Figures 4, 5: Please display the y=x (1:1) line for comparison. Grid lines could also make the figures easier to read.

To determine the accuracy of HydroColor vs. Aquafast, the statistics (R², median % error, mean square error) should be calculated point-by-point on the actual data, not on the regression line. Otherwise, you bias the results by introducing degrees of freedom (namely the regression line slope/intercept) and do not measure the actual accuracy.

Considering the HydroColor turbidity measurements have a typical uncertainty of 2.79 NTU (version 1) or 0.92 NTU (version 2), and the benchtop turbidity measurements have a typical uncertainty of 0.55 NTU, is the regression intercept of 2.6 NTU actually statistically significant?

361: “The update of [the] app” (typo).

What does the 1:2 relationship between Aquafluor IVChl and extracted Chl mean in practice? For example, if participatory science methods are being used to validate satellite data, should they use 2× the Aquafluor IVChl as the validation value for Chl? Or something else?

For the fluorescence data, a point-by-point 1:1 comparison (as discussed above) does not make sense because there is very clearly not a 1:1 relationship. Instead, I would suggest using the regression to rescale the data and then calculating the point-by-point statistics. See Figure 11 in reference [38] for an example.

Figure 6: It is very difficult to see the regression line between all the points. Please make it stand out, e.g. with colour or transparency. It would also be good to have it reach the vertical axis, to make the intercept visible.

The user-specific uncertainty analysis is very interesting. How much difference was there between users? Citizen science projects often consist of a few “super-users” who perform many measurements with high precision, and a lot of regular users who perform a handful of measurements with mixed precision. Did you observe a similar pattern?

524: “inherently the less replicate variability” - remove “the” (typo).

525: “1.5% of the median value” - which median value? Are you comparing the RMSD of all measurements to the median of all measurements? Or is it the median of the relative uncertainties? The latter would be more meaningful.

Might it be a good idea to restructure the Results section so that it is in order of measurement method (HydroColor validation → HydroColor variability → Fluorescence validation → Fluorescence variability)?

Discussion and conclusions

My only comment is that this discussion section is exemplary - very clearly written, detailed and complete but also concise and direct.

References

24: “Woerd HJvd” should be “van der Woerd HJ” (typo).

40: There is a space before the colon in the article title (typo).

Reviewer #2: An Interesting article discussing the efficacy of engaging citizens in monotoring of upper reaches of Chesapeake Bay. I understand the complexity asoxciated with measuring turbidity for the tributories that recdeived treated wasterwater or the shallow areas that are expsosed wave resuspension. How about the guidelines for measuremtns to accommodate boat wakes. Those shallow zones, particularly along the coastal areas, Boat wakes also cause significant respunspension of accumulated silt and clay and that could also interfre with the Hydrocolor measurement.

Also, one more variable is also need to consider, which is the rainfall and runoff from the watwershed in the aftermath of rainfall. That could also ater the CDOM and turbidity measurments.

Statistics and error estiamtes werre well presented and thisis a model that can be implemented for other Lakes and creeks around the country.

Some ninor suggestions

1. defenition of secchi disk can be improved.

6. PLOS authors have the option to publish the peer review history of their article (what does this mean?). If published, this will include your full peer review and any attached files.

Reviewer #1: No

Reviewer #2: No

---

## [Author Response · Author response to Decision Letter 0]

21 Aug 2024

Point-by-point responses (formatted version in included response to reviewer's file), line numbers refer to the tracked version:

Editor Comments:

1. Please ensure that your manuscript meets PLoS ONE's style requirements, including those for file naming. 

Response: These guidelines were followed in the preparation of the revisions.

Response: A sentence was added to methods (line 164): “No permits were required as sampling took place at sites with public access, or by volunteers sampling on their own property.”

3. We note that you have included the phrase “data not shown” in your manuscript…

Response: References to data not shown were removed from the manuscript. An alternate citation was provided.

4. We note that Figures 1 and 3 in your submission contain [map/satellite] images which may be copyrighted.

Response: A letter from the ESRI granting permission to use the map images is included in the submission files. Text stating that permission has been obtained and text citing original copyright holder is now included in captions of Fig. 1 and Fig. 3.

Reviewer Comments:

Reviewer #1:

This manuscript describes a set of participatory (citizen science) measurements of water quality parameters (turbidity, Chl, CDOM) in the Chesapeake Bay area. The low-cost participatory methods are compared to benchmark instruments to determine their accuracy, and replicate measurements are used to estimate the uncertainty.

The manuscript provides an interesting case study in using low-cost methods for water quality measurements, which would be interesting to anyone intending to perform similar measurements or use the resulting data. It is very well-written and easy to read. The discussion section in particular is excellent. However, there are significant shortcomings in the statistical analysis and data visualization, which are discussed in detail below, along with some minor revisions (mostly typographical). These major issues need to be resolved before the manuscript can be recommended for publication.

Response: We appreciate your positive feedback on the submission and have revised the manuscript to address your comments.

Abstract

The comparison between HydroColor v1 and v2 is very interesting to anyone who has used the app or is considering doing so in the future, so mentioning it in the abstract would increase the appeal of the manuscript.

Response: Abstract revised to mention the Hydrocolor update (line 38):

“Updates implemented in a new version (v2) of Hydrocolor improved the precision of estimates.”

Introduction

47, 48: For clarity, please specify that Chl and CDOM are concentrations.

Response: Added to the sentence to explain the quantities involved (line 50-51):

“These include biological indicators like chlorophyll a (Chl), physical indicators like turbidity, and chemical indicators like colored dissolved organic matter (CDOM), which relate, respectively, to the concentration of algae, suspended particles and organic substances in the water [1].”

110: “in vivo” is italicised here but not elsewhere.

Response: Removed italicization (line 115).

Methods

Figure 1 is rather low quality/DPI, making it difficult to read. Please replace it with a better quality image.

Response: A better resolution version has been provided for review, the full resolution version can be downloaded using a link that is provided at the top of the page of each figure.

178: There should be a comma after the closing bracket (typo).

Response: Corrected (line 187)

194-199: It would be good to clarify here that the satellite comparison is discussed in an upcoming manuscript, not in the present one. I kept expecting to see it while reading, but it only came back towards the end of the discussion as future work. Similarly, I’m not sure if “and ground truth remote sensing” really belongs in the title of this manuscript.

Response: Text that effect is included on lines 147-149:

“A future manuscript will focus on the optimization of algorithms used to interpret satellite remote sensing image data in Chesapeake Bay tributaries based on match-ups with volunteer measurements.”

The phrase “and ground truth remote sensing” is appropriate for the title because the variables chosen and the time & place of sampling has been selected to provide results that can be paired with satellite observations.

220: A 1:1 proportion, i.e. 1 NTU = 1 mg/L, or a linear relationship?

Response: It is correct as written, Leeuw et al (ref 21) assumed a 1:1 proportion in their validation of the Hydrocolor turbidity model.

226: Reference [38] would be appropriate here, since it was the basis for upgrading 

HydroColor to RAW imagery. It is very nice to see the authors testing the performance of the new app version in this manuscript.

Response: We agree, reference added, after revision it is reference # 26 (line 235)

263-264: What was the result of this daily check? How stable were the instrument readings?

Response: Details on the daily check were added to the text (line 274-276):

“The instrument was checked daily before use that readings were within 1 NTU of a 20 NTU standard and deionized water for 0 NTU. Instruments rarely failed the check, which if it happened was usually due to contamination on the sample vial. “

269: How were these cases selected? Could the selection process have introduced a bias in the results?

Response: Details were added about sample selection for validation (line 282-284):

“Sample selection was mainly driven by the logistical concerns of timely transport and processing, but care was taken to acquire samples from all water bodies and in all seasons.” 

To address the issue of whether the selection process introduced bias, we have modified Table 1 to include the quartiles and mean absolute deviations of replicates for the volunteer samples selected for validation. Text has been added that describes this addition to Table 1 (lines 318-319):

“Samples acquired for validation had similar characteristics as the whole data set, shifted to slightly higher values (Table 1).”

273: There is a space between the ~ and 50 (typo)

Response: Corrected (line 286).

The aggregate RSMD (Equation 2) is not a sufficient description of the uncertainty/variability in the volunteer measurements. It only provides a very rough measure of the variability of one method, but is sensitive to outliers, depends on the unit scale, and cannot be used in further analysis, e.g. weighted regression. It would be better to estimate the uncertainty in each measurement from its replicates (the inner sum in Equation 2) and plot these in the scatter plots (if visually possible) and use them to weight the regression line and statistics. To express the variability of each method, present the distribution of these individual uncertainty estimates, e.g. as a histogram or boxplot, or simply with the me(di)an and some chosen quantiles (1 or 2 σ, 25%-75%, etc.). The me(di)an uncertainty can then be used in the rest of the analysis, e.g. Tables 1 and 2. It would also be very useful to know if the uncertainty depends on the value - the same absolute uncertainty in NTU has a very different significance to a 5 NTU measurement vs. a 30 NTU measurement.

Response: We agree with the reviewer that RMSD is sensitive to outliers so have revised the replicate variation statistic to be the mean absolute deviation (MAD). These are now shown in Table 1. In general, though, whether summarized as RMSD or MAD, replicate variation is relatively small. These results demonstrate that volunteers produce consistent observations using project methods.

Results

Table 1 shows fluorescence first, then turbidity, while the rest of the manuscript discusses them in the opposite order. It would be more intuitive to keep the order consistent.

Response: Good suggestion, the table order was changed so turbidity is listed first. The order of panels in Figure 2 was also changed to have turbidity at the top. Regression results have been moved to a new Table 2.

Figure 2: The trend line should be removed during the periods without data, approximately November-February in panels (a) and (b). Moreover, the caption should explain what type of average the trend line is and what its window width is.

Response: Changes made as suggested. Caption explains that trend line was calculated using locally weighted regression (LOESS).

Figure 2: I’m not sure I understand why the high-CDOM outlier was removed. Doesn’t it further prove the point that short-term variability is very high, thus making a case for participatory monitoring? Or is it because the pollen do not result in actual CDOM in the water, but disturb the measurement? In the latter case, the caption should be changed to clarify this, as it currently suggests the CDOM concentration itself was high. Furthermore, was this effect negligible on all other days?

Response: The point was omitted from the plot and calculation of the trend line since it only occurred at one site. The CDOM measurement is real, it was just highly localized to a site where wind patterns concentrated pollen into one location. To make it clear when this event occurred, an indicating arrow was added to the plot and text added to the results line 349-351:

“Measurements at the three docks were generally consistent, except for April 6, 2022 when there was a high accumulation of tree pollen at one site which lead to an unusually high CDOM reading (Figure 2b) compared to the other sites.”

Figures 2, 4, 5, 6: These figures would benefit from some kind of visual indicator of the uncertainty in the data, so the reader can determine the significance of outliers/differences without having to go back and forth between figures and tables. If the uncertainties differ a lot between points, then consider plotting individual bars; if not, then consider plotting an example in one of the empty corners of each panel.

Response: As shown by the listed MAD, the replicate variability in the volunteer data is small, in most cases an error bar indicating sample specific MAD would not extend beyond the symbol. There was no indication that replicate variability was substantially greater in the upper range of observations.

Figures 2, 4, 5, 6: Additional panels showing the residuals (y-x or regression-value) would make it clear if there are any biases, how the error depends on the mean value, etc.

Response: Plots of the residuals for Figures 4,5,6 are included in the supporting information, S2 Fig. We thank the reviewer for that suggestion as these plots did reveal a bias in estimating the AQUAfast turbidity on the basis of HydroColor if the intercept was ignored. The turbidity regressions now incorporate intercepts and the residuals are evenly distributed around zero. Tables and text have been updated with revised regression results.

Figure 3: This has the same image quality/DPI issue as Figure 1.

Response: A higher resolution version is provided for the proof, use the link at the top of the page to download the full resolution image.

Figure 3: Please change the colour bar to be more representative of the results and easier to read for those with colour blindness. See [https://doi.org/10.1038/s41467-020-19160-7]. Between this and the low DPI, it is quite difficult to see the suggested upstream/downstream effect. A thin black outline around each point would also help - there are some spots where I’m not sure if I’m seeing a dot or a bend in the coastline.

Response: The color bar on the figure has been recoded to the “viridis” scale, which is one of the recommended color sets. Outlines have been added to points to assist in distinguishing them from the map and from other points. 

Figure 4: It would make more sense to have the benchmark measurements on the horizontal axis and the HydroColor measurements on the vertical axis, since the latter is being validated to the former.

Response: All of the volunteer measurements are essentially proxies for the variables of interest, so there is no expectation of a 1:1 relationship as there might be, for example, in comparing remote sensing reflectance (Rrs) between Hydrocolor and a research spectrometer. The purpose of the validation comparisons is to determine the best fit linear relationship between the target variable and the proxy, in which case, it is appropriate for the proxy to be the independent variable. Although Hydrocolor measurements are read off in NTUs, they are derived by an assumed relationship between Rrs and turbidity. As will be seen in our forthcoming submission on the remote sensing algorithms, the fitted slopes are used to reparamaterized volunteer data for use in ground truthing. This objective is now emphasized in text added to the introduction (lines 145-147, new text in red):

 “This manuscript covers the methods used, the data acquired over two years of deployment, and shows that after reparametrizing on the basis of validation data, volunteer data is suitable for ground truthing remotely-sensed water quality.”

Figure 4: Very minor - you could add (v1) and (v2) to the panel titles to make it more immediately obvious what the difference between the two is.

Response: Version labels added to plots

Figure 4: The difference in spread would be more visible if both panels used the same scale/limits on the horizontal and vertical axes.

Response: Axis limits changed to be the same in each plot

Figures 4, 5: Please display the y=x (1:1) line for comparison. Grid lines could also make the figures easier to read.

Response: Since the volunteer measurements are proxies there is no expectation of a 1:1 relationship. We did add grid lines to aid in reading the figures.

To determine the accuracy of HydroColor vs. AQUAfast, the statistics (R², median % error, mean square error) should be calculated point-by-point on the actual data, not on the regression line. Otherwise, you bias the results by introducing degrees of freedom (namely the regression line slope/intercept) and do not measure the actual accuracy.

Response: As mentioned above, the regression line should be considered the basis of comparison.

Considering the HydroColor turbidity measurements have a typical uncertainty of 2.79 NTU (version 1) or 0.92 NTU (version 2), and the benchtop turbidity measurements have a typical uncertainty of 0.55 NTU, is the regression intercept of 2.6 NTU actually statistically significant?

Response: With n =~200, the intercept is significant (p < 0.001). In this revision, the validation regressions include the intercept. 

361: “The update of [the] app” (typo).

What does the 1:2 relationship between Aquafluor IVChl and extracted Chl mean in practice? For example, if participatory science methods are being used to validate satellite data, should they use 2× the Aquafluor IVChl as the validation value for Chl? Or something else?

Response: Correct, this is the intention of the study which is hopefully clear with the text added to lines 145-147.

For the fluorescence data, a point-by-point 1:1 comparison (as discussed above) does not make sense because there is very clearly not a 1:1 relationship. Instead, I would suggest using the regression to rescale the data and then calculating the point-by-point statistics. See Figure 11 in reference [38] for an example.

Response: The RMSE is mathematically identical to the calculation the reviewer describes.

Figure 6: It is very difficult to see the regression line between all the points. Please make it stand out, e.g. with colour or transparency. It would also be good to have it reach the vertical axis, to make the intercept visible.

Response: The regression line is colored blue to stand out among the points.

The user-specific uncertainty analysis is very interesting. How much difference was there between users? Citizen science projects often consist of a few “super-users” who perform many measurements with high precision, and a lot of regular users who perform a handful of measurements with mixed pr

---

## [Decision Letter · Decision Letter 1]

20 Sep 2024

Participatory science methods to monitor water quality and ground truth remote sensing of the Chesapeake Bay

PONE-D-24-22133R1

Dear Dr. Neale,

We’re pleased to inform you that your manuscript has been judged scientifically suitable for publication and will be formally accepted for publication once it meets all outstanding technical requirements.

Kind regards,

Steven Arthur Loiselle

Academic Editor

PLOS ONE

Additional Editor Comments (optional):

Reviewers' comments:

Reviewer's Responses to Questions

**Comments to the Author**

1. If the authors have adequately addressed your comments raised in a previous round of review and you feel that this manuscript is now acceptable for publication, you may indicate that here to bypass the “Comments to the Author” section, enter your conflict of interest statement in the “Confidential to Editor” section, and submit your "Accept" recommendation.

Reviewer #1: All comments have been addressed

2. Is the manuscript technically sound, and do the data support the conclusions?

Reviewer #1: Yes

3. Has the statistical analysis been performed appropriately and rigorously? 

Reviewer #1: Yes

4. Have the authors made all data underlying the findings in their manuscript fully available?

Reviewer #1: Yes

5. Is the manuscript presented in an intelligible fashion and written in standard English?

Reviewer #1: Yes

6. Review Comments to the Author

Reviewer #1: (No Response)

7. PLOS authors have the option to publish the peer review history of their article (what does this mean?). If published, this will include your full peer review and any attached files.

Reviewer #1: No

---

## [Editor Report · Acceptance letter]

9 Oct 2024

PONE-D-24-22133R1 

PLOS ONE

Dear Dr. Neale, 

I'm pleased to inform you that your manuscript has been deemed suitable for publication in PLOS ONE. Congratulations! Your manuscript is now being handed over to our production team.

Kind regards, 

on behalf of

Dr. Steven Arthur Loiselle 

Academic Editor

PLOS ONE